# Spectral structure learning for clinical time series

Ivan Lerner[1,2], Anita Burgun[1,2], and Francis Bach[3]

[1]*Inserm, Centre de Recherche des Cordeliers, Université Paris Cité, Sorbonne Université, F-75006 Paris, France*
[2]*HeKA, Inria Paris, F-75012 Paris,France*
[3]*SIERRA, Inria Paris, F-75015 Paris, France*

## Abstract

We develop and evaluate a structure learning algorithm for clinical time series. Clinical time series are multivariate time series observed in multiple patients and irregularly sampled, challenging existing structure learning algorithms. We assume that our times series are realizations of StructGP, a $k$-dimensional multi-output or multi-task stationary Gaussian process (GP), with independent patients sharing the same covariance function. StructGP encodes ordered conditional relations between time series, represented in a directed acyclic graph. We implement an adapted NOTEARS algorithm, which based on a differentiable definition of acyclicity, recovers the graph by solving a series of continuous optimization problems. Simulation results show that up to mean degree 3 and 20 tasks, we reach a median recall of 0.93% [IQR, 0.86, 0.97] while keeping a median precision of 0.71% [0.57-0.84], for recovering directed edges. We further show that the regularization path is key to identifying the graph. With StructGP, we proposed a model of time series dependencies, that flexibly adapt to different time series regularity, while enabling us to learn these dependencies from observations.

## 1 Introduction

Structure learning is the task of learning the dependency structure of either time-independent variables or, in this study, time series [1, 2]. The structure of dependency is usually represented as a directed acyclic graph (DAG), in which nodes represent variables, and links between nodes represent relations between variables. These links encode conditional or marginal independence relations [3, 4], or under certain strong additional assumptions (e.g., no hidden cofounders), have a causal interpretation and the structure can be used for *causal modeling* [1, Chapter 6].

Structure learning algorithms are typically classified into constraint-based methods and score-based methods [2]. Score-based methods assume a parametric model in which the parameters support the graph, and then search for the graph that best fits the data [5]. The main limitation of this approach is that the acyclicity of the graph requires an iterative search over graph structures that satisfy the constraint. And that the number of acyclic graphs grows superexponentially with the number of nodes in the graph [6]. Formulating the acyclicity of a directed acyclic graph (DAG) as a differentiable function of its adjacency matrix allowed Zheng et al. [7] to frame the problem as a continuous optimization problem. Following this line of work, DYNOTEARS [8] uses the acyclicity constraint to identify the structure of linear structural vector autoregressive models (SVAR) [9].

In this study, we aim to develop a structure learning algorithm for clinical time series collected from Electronic Health Records (EHRs). EHRs time series are collections of irregularly sampled multivariate time series, collected in multiple patients. To learn graphical models of dependence between those, we work in continuous time and develop StructGP, a structured multi-task Gaussian process (GP) [10].

Workshop on Bayesian Decision-making and Uncertainty, 38th Conference on Neural Information Processing Systems (NeurIPS 2024).

## 2 Methods

### 2.1 Structured Gaussian process

We consider the $k$-dimensional multi-output Gaussian process $\mathbf{Y}(t)$ defined as the filtration by $\mathbf{H}(t)$ of white noise $\mathbf{w}(t)$:

$$\mathbf{Y}(t) = (\mathbf{H} * \mathbf{w})(t),$$

where $\mathbf{H}(t)$ is a sparse $k \times k$ lower triangular matrix-valued impulse response function, and $\mathbf{w}(t)$ is a $k$-dimensional white noise vector. Taking the Fourier transform of $\mathbf{Y}(t)$, we find $\mathbf{Z}(\omega)$ a non-stationary complex white noise multi-output process [11, p. 418]:

$$\mathbf{Z}(\omega) = \tilde{\mathbf{H}}(\omega)\mathbf{W}(\omega), \tag{1}$$

where $\tilde{\mathbf{H}}(\omega)$ is the Fourier transform of $\mathbf{H}(t)$, and $\mathbf{W}(\omega)$ complex independent white noise processes. In addition, the covariance of $\mathbf{Z}(\omega)$ or *spectral density* is:

$$\mathrm{Cov}\left(\mathbf{Z}(\omega), \mathbf{Z}(\omega')\right) = \begin{cases} \tilde{\mathbf{H}}(\omega)\tilde{\mathbf{H}}^T(\omega) & \text{if } \omega = \omega' \\ 0 & \text{if } \omega \neq \omega' \end{cases}.$$

Thus, it identify $\tilde{\mathbf{H}}(\omega)$ as the Cholesky factor of the covariance matrix of $\mathbf{Z}(\omega)$. Given Equation 6 in Appendix A6.3, we find that the sparsity pattern of $\mathbf{H}$ parameterizes ordered conditional relations between time series:

$$\mathbf{H}_{vu}(t) = 0, \quad \forall t \in \mathbb{R}$$
$$\Leftrightarrow \quad \tilde{\mathbf{H}}_{vu}(\omega) = 0, \quad \forall \omega \in (-\pi, \pi)$$
$$\Leftrightarrow \quad Z_u \perp\!\!\!\perp Z_v \mid \mathbf{Z}_{\{1,2,\ldots,u-1\}}$$
$$\Leftrightarrow \quad \mathbf{f}_{uv|C}(\omega) = \mathbf{f}_{uv}(\omega) - \mathbf{f}_{uC}(\omega)\mathbf{f}_{CC}^{-1}(\omega)\mathbf{f}_{Cv}(\omega) = 0, \quad \forall \omega \in (-\pi, \pi)$$
$$\Leftrightarrow \quad Y_u \perp\!\!\!\perp Y_v \mid \mathbf{Y}_{\{1,2,\ldots,u-1\}},$$

where $\mathbf{f}_{uv|C}$ is the partial cross spectrum of $\mathbf{Y}_u$ and $\mathbf{Y}_v$ given $C = \mathbf{Y}_{\{1,2,\ldots,u-1\}}$.

We therefore parameterize $\mathbf{H}(t)$ as follows:

$$\mathbf{H}(t) = (\mathbf{I} - \mathbf{S}) \circ \mathbf{L}(t), \tag{2}$$

where $\mathbf{L}_{vu}(t) = \exp\left(-\frac{t^2}{\mathbf{L}_{vu}}\right)$, $\mathbf{S}$ is a sparse lower triangular matrix up to permutation, $\mathbf{L}$ is a positive square matrix, $\mathbf{I}$ is the identity matrix, and $\circ$ the Hadamard product. The support of $\mathbf{S}$ can then be interpreted as the adjacency matrix of a directed acyclic graph (DAG) $\mathcal{G}$ that encodes ordered conditional independence relation:

$$\mathbf{S}_{vu} \neq 0 \tag{3}$$
$$\Leftrightarrow Y_u \xrightarrow{\mathcal{G}} Y_v$$
$$\Leftrightarrow Y_u \perp\!\!\!\perp Y_v \mid \mathbf{Y}_{\{1,2,\ldots,u-1\}}.$$

And the distribution $P(\mathbf{Y})$ satisfy the Markov factorization property with respect to the graph $\mathcal{G}$ [4, Theorem 2.49]:

$$P(\mathbf{Y}) = \prod_{u=1}^{k} P\big(Y_u \mid \mathrm{pa}(Y_u)\big). \tag{4}$$

### 2.2 Learning the graph

As our time series are irregularly sampled from multiple patients we switch here to a set-based indexing, thanks to the marginalization properties of GP. Our data is then a collection of $n$ scalar observations $y$ from $r$ individuals and $k$ tasks, $\{y(\mathbf{x}) : \mathbf{x} \in (\mathbb{N}, \mathbb{N}, \mathbb{R}^+)\}$. Each observation is indexed by the input vector $\mathbf{x}$, a triplet composed of the patient index $i$, task index $j$, and time $t$, such that $\mathbf{x} = (i, j, t)$, with $i \in \{1, \ldots, r\}$, $j \in \{1, \ldots, k\}$ and $t \in \mathbf{R}^+$. We observe multiple independent patients, and the intra-patient covariance of the process for patient $i$ can be written:

$$\mathrm{Cov}[\mathbf{Y}(i, u, t), \mathbf{Y}(i, v, t')] = (\mathbf{h}_u * \mathbf{h}_v^T)(t - t').$$

With classical GP, the set of free parameters $\theta = \{\mathbf{S}, \mathbf{L}\}$ is learned by maximizing $\log p(\mathbf{y}|\mathbf{X}, \theta)$, the marginal likelihood of the training observations $\mathbf{y}$ given inputs $\mathbf{X}$ [12]. With structured GP, we follow the NOTEARS algorithm to learn the graph, order of tasks, and sparsity pattern [7] and impose an acyclicity constraint on $\mathbf{S}$. NOTEARS leverages the trace of the matrix exponential of the adjacency matrix as a differentiable acyclicity constraint. Our objective is therefore to solve the constrained optimization problem below:

$$\theta^* = \operatorname*{argmin}_{\theta} - \log p(\mathbf{y}, \mathbf{X}, \theta) + \lambda \|\mathbf{S}\|_1 \qquad (5)$$
$$\textit{s.t.} \ \ \mathrm{Tr}(\exp(\mathbf{S} \circ \mathbf{S})) - k = 0,$$

where $\lambda$ is the penalty strength.

The above is solved by dual ascent following the augmented Lagrangian method [13], such that the constrained problem is equivalent to solving a series of unconstrained problems, the primal and the dual. The primal is the penalized objective function augmented with a quadratic penalty term, and is solved with a proximal gradient method (see Appendix A6.5), the dual is solved by gradient ascent (see Appendix A6.4). Finally, we find by grid search $\lambda_*$, the sparsity penalty that minimizes an equivalent of the Akaike information criterion (AIC): $\mathrm{AIC} = 2\|\mathbf{S}\|_0 - 2 \log \mathcal{L}(\mathbf{y}, \mathbf{X}, \theta)$. Grid search is conducted from $\lambda_{max}$ to $\lambda_{min}$ on a log-scale, with warm-start. Because solving the augmented Lagrangian problem to small error is computationally expensive, and leads to numerical instability when $\rho$ becomes large, we choose to only loosely solve it, typically with a large tolerance for the acyclicity constraint ($\epsilon = 0.1$, see Appendix A6.4). Thus, to ensure dagness, we apply a hard-threshold operation, i.e. we mask elements in $\mathbf{S}$ lower than the minimal threshold that ensures dagness.

## 2.3 Simulation study

| Experiment | Number of tasks | Mean degree | Grid search steps | Number of patients |
|---|---|---|---|---|
| | $k$ | $md$ | $n_\lambda$ | $r$ |
| TOY | 4 | 2 | 256 | 50 |
| EXP1 | 10 | 2 | 50 | $[1, \ldots, 100]$ |
| EXP2 | 10 | 3 | $[2, \ldots, 512]$ | 50 |
| EXP3 | $[2, \ldots, 20]$ | $[1, 2, 3]$ | 50 | 50 |

Table 1: Summary of simulation parameters

We empirically assess the accuracy of the algorithm to identify a graph from observations through a series of simulations. For each simulation, we sample a graph, the covariance parameters $\theta$, and observations from the sampled prior GP. The definition of the GP follows that of section 2.1, with the additional constraint that lengthscales parameters are tied for each task and exponentiated ( $\mathbf{L}_{vu} = \exp(\ell_v)$ for all $v \in \{1, 2, \ldots, k\}$, $u \in \{1, 2, \ldots, k\}$). We then fit the model using the overall algorithm from section 2.2 (including the grid search), and compare the predicted graph $\hat{\mathcal{G}}$ with the true simulated graph $\mathcal{G}$. The comparison is made with the structural hamming distance (SHD). SHD is the "edit distance" of graphs, it counts the number of edge modifications (insertion, deletion, inversion) necessary to transform a predicted graph into the true simulated graph. We also compare the root mean square error (RMSE) between the predicted $\hat{\mathbf{S}}$ and true (simulated) $\mathbf{S}$ parameters. Each simulation is repeated 100 times and we report the average metric along with its bootstrapped 95% confidence intervals. For comparison purposes, we also report the same metrics for a random graph from the same distribution as simulated.

In all simulations, the support of $\mathbf{S}$ is sampled from a random (Erdős–Rényi) graph in which the sparsity level is controlled by the mean degree of the graph $md$. $\mathbf{S}$ parameters are uniformly sampled in $[-2, -0.5] \cup [0.5, 2]$. $\ell_u$ parameters are uniformly sampled in $[-0.5, 0.5]$. The observation times, $t$, are uniformly sampled in $[0, 10]$. The observation noise level is fixed at $\sigma = 0.01$ and given as oracle when learning. We first report results from one simulation 'TOY', a toy model with 4 tasks that illustrate how to compute counterfactual trajectories and how we recover the graph from observations. We then report 3 experiments each varying specific parameters (see Table 1), while the number of observations per task is fixed at ($n_k = 10$). Code available at gitlab.

# 3 Results

## 3.1 Toy model

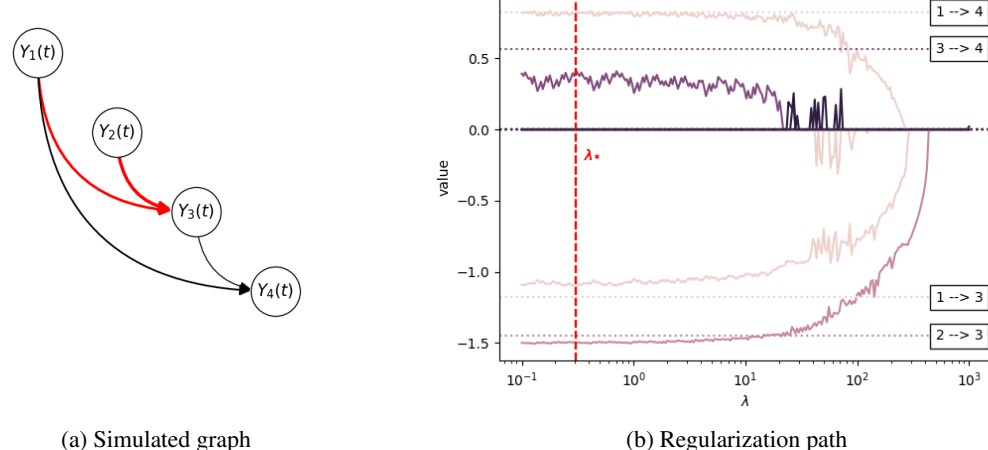

(a) Simulated graph             (b) Regularization path

Figure 1: A toy model with 4 tasks
We sample a random graph whose adjacency matrix sparsity pattern encodes ordered conditional relations between time series (a). We recover the parameters and order of variables from observations sampled uniformly and independently between tasks (see section 2.3).

In Figure 1a, we show a sampled random DAG of mean degree 2 for 4 tasks, with 4 links. A link can be interpreted as the presence of a direct or indirect effect. It corresponds to the following output scale parameters of the impulse response function $\mathbf{H}(t)$:

$$\mathbf{I} - \mathbf{S} = \begin{bmatrix} 1 & 0 & 0 & 0 \\ 0 & 1 & 0 & 0 \\ -1.18 & -1.45 & 1 & 0 \\ 0.82 & 0 & 0.57 & 1 \end{bmatrix}.$$

This graph encodes the following ordered independence relations:

$$Y_4 \perp\!\!\!\perp Y_2 \mid Y_1,$$
$$Y_2 \perp\!\!\!\perp Y_1 \mid \varnothing.$$

Which corresponds to the following Markov factorization of the distribution:

$$P(\mathbf{Y}) = P(Y_4 \mid Y_3, Y_2, Y_1)P(Y_3 \mid Y_2, Y_1)P(Y_2 \mid Y_1)P(Y_1)$$
$$= P(Y_4 \mid Y_3, Y_1)P(Y_3 \mid Y_2, Y_1)P(Y_2)P(Y_1).$$

Figure 1b shows that with 50 patients and 10 observations per task, our algorithm exactly recovers the order of variables and the sparsity pattern. The learned weights are slightly biased toward 0 due to the regularization properties of our objective function, the *nmll*.

## 3.2 Simulation study

We report in Figure 2 the results of 'EXP1' which shows that the algorithm from section 2.2 identifies the true graph almost perfectly in this 'ideal' regime of parameters (large weights, sparse graphs, low noise). Indeed, with 100 patients, most of the errors that contribute to the SHD are extra links. We then show in supplementary Figure S3, that the number of grid search steps is heavily impacting the capacity to recover the true graph (red dots), but not because it allows us to precisely infer the optimal $\lambda_*$ (green dots). In fact, fitting the model from a random initialization with the optimal $\lambda_*$ produces very poor performances, and improving the number of grid search steps does not improve the solutions (green dots). In supplementary Figure S4, we see that up to mean degree 3 and 20 tasks,

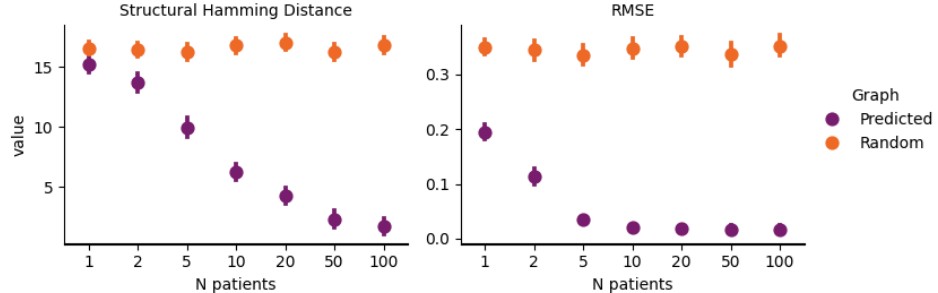

Figure 2: Average metrics for an increasing number of patients
Reports of 'EXP1', increasing the number of patients for 10 tasks and 10 observations per task. The predicted graph (purple dots) is compared with a random graph from the same graph distribution (orange dots). The simulated graphs are random graphs with mean degree 2. Error bars represent bootstrap 95% confidence intervals.

most errors are spurious links. Table 2 shows that, for recovering directed edges, we reach a median recall of 0.93% [IQR, 0.86, 0.97] while keeping a median precision of 0.71% [0.57-0.84] on 20 nodes graphs.

| $k$ | **P** (Predicted) | **P** (Random) | **R** (Predicted) | **R** (Random) |
|---|---|---|---|---|
| 4 | 1.00 [0.83-1.00] | 0.50 [0.33-0.67] | 1.00 [0.83-1.00] | 0.50 [0.33-0.67] |
| 10 | 0.82 [0.69-0.93] | 0.14 [0.08-0.25] | 0.94 [0.87-1.00] | 0.16 [0.07-0.25] |
| 20 | 0.71 [0.57-0.84] | 0.07 [0.04-0.11] | 0.93 [0.86-0.97] | 0.07 [0.04-0.12] |

Table 2: Precision and recall
Precision (**P**) and recall (**R**) median and interquartile for varying mean degree and number of task parameters, from 100 replications of 'EXP3' simulations with $md = 3$. False positives include extra links and reversed links. Metrics are computed from models learned (Predicted) and compared with metrics for random graphs of the same distribution as the simulated data (Random).

## 4 Discussion

We developed a structure learning algorithm that learns ordered conditional independence relations between irregularly sampled time series. These relations are parametrized by StructGP, a GP model built upon the convolution between a sparse lower-triangular matrix-valued impulse response function and independent white noises. It corresponds to assuming a linear additive Gaussian SCM for the Fourier representation of the time series, whose structure is invariant over all frequencies. Based on a differentiable definition of acyclicity, this algorithm recovers the true graph by solving a series of continuous optimization problems with high sensitivity and good precision on simulated data.

The recent work by Dallakyan [14] is the closest to ours. They develop a structure learning algorithm for time series by imposing a Gaussian linear additive SCM on the discrete Fourier transform of the time series. However, they learn different weight matrices at each frequency, whereas we assume an invariant structure across continuous frequencies, parametrized by only one weight matrix.

More work will be needed to bridge the gap between simulated data and real-world data with regard to the sensitivity to standardization and unmeasured cofounders. Indeed, when standardizing the time series, it was reported that with time-independent variables, standardization affected graph recovery [15, 16]. However, it is possible to re-parameterise the SCM to ensure a marginal unit variance without losing the identifiability of the graph from observations [17]. Furthermore, scaling to large datasets could be achieved, for instance, with GPU-friendly solvers that exploit block sparsity induced by independence between patients [18].

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

## 5  List of abbreviations

**DAG**  directed acyclic graph. 1, 2, 4

**EHRs**  Electronic Health Records. 1

**GP**  Gaussian process. 1–3, 5

*nmll*  negative marginal log-likelihood. 4

**NOTEARS**  Non-combinatorial Optimization via Trace Exponential and Augmented lagRangian for Structure learning. 3

**PGM**  proximal gradient method. 3, 9

**RMSE**  root mean square error. 3

**SCM**  structural causal model. 5

**SHD**  structural hamming distance. 3, 4

**StructGP**  Structured Gaussian process. 1, 5

**SVAR**  structural vector autoregressive models. 1

## 6  Appendix

### 6.1  Conditional independence

These sections adapt proofs from [4, 19] with the same notations as the remainder of the article for simplicity.

### 6.2  Sparsity pattern of the precision matrix encodes conditional independence

Let $\mathbf{Y} = (Y_1, Y_2, \ldots, Y_k)^\top$ be a multivariate normal random vector with mean vector $\mu$ and covariance matrix $K$, i.e., $\mathbf{Y} \sim \mathcal{N}(\mu, K)$.

Consider the two sets of variables $A = \{u, v\}$ and $B = \{1, 2, \ldots, k\} \backslash \{u, v\}$. The covariance matrix can be partitioned as:

$$K = \begin{pmatrix} \mathbf{K}_A & \mathbf{K}_{AB} \\ \mathbf{K}_{AB}^T & \mathbf{K}_B \end{pmatrix}.$$

After conditioning on $B$, the conditional covariance matrix is given by:

$$\mathbf{K}_{A|B} = \mathbf{K}_A - \mathbf{K}_{AB}\mathbf{K}_B^{-1}\mathbf{K}_{AB}^T.$$

Now, define the precision matrix $\boldsymbol{\Omega} = \mathbf{K}^{-1}$ with the same partition:

$$\boldsymbol{\Omega} = \begin{pmatrix} \boldsymbol{\Omega}_A & \boldsymbol{\Omega}_{AB} \\ \boldsymbol{\Omega}_{AB}^T & \boldsymbol{\Omega}_B \end{pmatrix}.$$

Using the inversion of a partitioned matrix, we recognize that:

$$\boldsymbol{\Omega}_A = (\mathbf{K}_A - \mathbf{K}_{AB}\mathbf{K}_B^{-1}\mathbf{K}_{AB}^T)^{-1}.$$

Or otherwise stated that the precision matrix is invariant to conditioning:

$$\mathbf{\Omega}_{A|B} = \mathbf{K}_{A|B}^{-1} = \mathbf{\Omega}_A.$$

Thus, let:

$$\mathbf{\Omega}_A = \begin{pmatrix} \omega_{uu} & \omega_{uv} \\ \omega_{uv} & \omega_{vv} \end{pmatrix}.$$

Then, we have:

$$\mathbf{K}_{A|B} = \frac{1}{\det(\mathbf{\Omega}_A)} \begin{pmatrix} \omega_{vv} & -\omega_{uv} \\ -\omega_{uv} & \omega_{uu} \end{pmatrix}.$$

This shows that:

$$Y_u \perp\!\!\!\perp Y_v \mid \mathbf{Y}_{\backslash\{u,v\}} \Leftrightarrow \omega_{uv} = 0.$$

## 6.3 Sparsity pattern of the Cholesky factor of the covariance matrix encodes ordered conditional independence

Let $\mathbf{Y} = (Y_1, Y_2, \ldots, Y_k)^\top$ be a multivariate normal random vector with mean vector $\mu$ and covariance matrix $\mathbf{K}$, i.e., $\mathbf{Y} \sim \mathcal{N}(\mu, \mathbf{K})$.

Consider the two sets of variables $A = \{1, 2, \ldots, u-1\}$ and $B = \{u, \ldots, k\}$. The covariance matrix $\mathbf{K}$ can be partitioned and factorized as follows:

$$\mathbf{K} = \begin{pmatrix} \mathbf{K}_A & \mathbf{K}_{AB} \\ \mathbf{K}_{AB}^T & \mathbf{K}_B \end{pmatrix}$$
$$= \begin{pmatrix} I & 0 \\ \mathbf{K}_{AB}^\top \mathbf{K}_A^{-1} & I \end{pmatrix} \begin{pmatrix} \mathbf{K}_A & 0 \\ 0 & \mathbf{K}_B - \mathbf{K}_{AB}^\top \mathbf{K}_A^{-1} \mathbf{K}_{AB} \end{pmatrix} \begin{pmatrix} I & \mathbf{K}_A^{-1} \mathbf{K}_{AB} \\ 0 & I \end{pmatrix}.$$

Hence, the Cholesky factor $\mathbf{L}$ of $\mathbf{K}$ is given by:

$$\mathbf{L} = \begin{pmatrix} \mathbf{L}_A & 0 \\ \mathbf{K}_{AB}^\top \mathbf{L}_A^{-1} & \mathbf{L}_S \end{pmatrix},$$
where
$$\mathbf{L}_A = \mathrm{chol}(\mathbf{K}_A),$$
$$\text{and } \mathbf{L}_S = \mathrm{chol}(\mathbf{K}_B - \mathbf{K}_{AB}^\top \mathbf{K}_A^{-1} \mathbf{K}_{AB}).$$

The element of the $v$-th row and $u$-th column, $\mathbf{L}_{vu}$, can be found in $\mathbf{L}_S$ in the first column of at the row $(v - u + 1)$:

$$\mathbf{L}_{vu} = \mathbf{L}_{S,(v-u+1)1}.$$

The conditional covariance given $A$ (i.e., all the variables preceding $u$) is:

$$\mathbf{K}_{B|A} = \mathbf{L}_S \mathbf{L}_S^T.$$

This shows that:

$$Y_u \perp\!\!\!\perp Y_v \mid \mathbf{Y}_{\{1,2,\ldots,u-1\}} \tag{6}$$
$$\Leftrightarrow \quad \mathbf{K}_{B|A,v1} = 0$$
$$\Leftrightarrow \quad \sum_{u=1}^{k} \mathbf{L}_{S,(v-u+1)u} \mathbf{L}_{S,1u} = 0$$
$$\Leftrightarrow \quad \mathbf{L}_{S,(v-u+1)1} = 0$$
$$\Leftrightarrow \quad \mathbf{L}_{vu} = 0.$$

## 6.4 The augmented Lagrangian optimization algorithm

Given:

- Objective function: $f(\theta) = -\log \mathcal{L}(\mathbf{y}, \mathbf{X}, \theta) + \mathcal{P}_\lambda(\mathbf{S})$
- Constraint: $g(\theta) = \mathbf{tr}(\exp(\mathbf{S} \circ \mathbf{S})) - k = 0$
- Convergence criteria: constraint tolerance $\epsilon$ and $\rho_{max}$
- Primal solver: *Solver*

Define:

- Lagrange multiplier $\alpha^{(k)}$ at step $k$
- Augmented Lagrangian: $L(\theta, \alpha^{(k)}, \rho) = f(\theta) + \alpha^{(k)} g(\theta) + \frac{\rho}{2} g(\theta)^2$

Do:

1. Choose initial guess $\theta^{(0)}$, Lagrange multipliers $\alpha^{(0)} = 0$ and $\rho = 1$
2. For $k = 0, 1, 2, \ldots$
   
   (a) While $\rho < \rho_{max}$ update $\theta^{(k+1)}$.
   - Minimize the augmented Lagrangian function:
   $$\theta^{(k+1)} = Solver\big(L(\theta, \alpha^{(k)}, \rho)\big)$$
   - Break if $g(.)$ sufficiently decreases:
   $$g(\theta^{(k+1)}) < 0.25\, g(\theta^{(k)})$$
   - Else augment $\rho$:
   $$\rho = 10\, \rho$$
   
   (b) Update Lagrange multipliers:
   $$\alpha^{(k+1)} = \alpha^{(k)} + \rho \cdot g(\theta^{(k+1)})$$
   
   (c) Check convergence criteria. If satisfied, stop.
   $$g(\theta^{(k+1)}) < \epsilon \text{ or } \rho \geqslant \rho_{max}$$

## 6.5 Proximal gradient method

PGM [20] is a generalization of gradient descent for objective functions that can be split between a differentiable and non-differentiable part. It requires defining a proximal operator for the non-differentiable part:

$$\mathbf{prox}_\lambda(\mathbf{S})_{uv} = \begin{cases} s_{uv} - \lambda, & \text{if } s_{uv} > \lambda \\ 0, & \text{if } s_{uv} \leqslant |\lambda| \\ s_{uv} + \lambda, & \text{if } s_{uv} < -\lambda. \end{cases}$$

Parameters are found by taking a classic gradient step followed by a proximal step at each iteration $k$:

$$\text{(i) } \theta^{k+\frac{1}{2}} = \theta^k - \alpha^k \nabla(-\log \mathcal{L}(\mathbf{y}, \mathbf{X}, \theta^k))$$
$$\text{(ii) } \theta^{k+1} = \mathbf{prox}_{\alpha^k \lambda}(\theta^{k+\frac{1}{2}})$$

with $\alpha^k$ a learning rate determined by line search at each iteration, following Beck and Teboulle [21].

## 6.6   Simulation study additional results

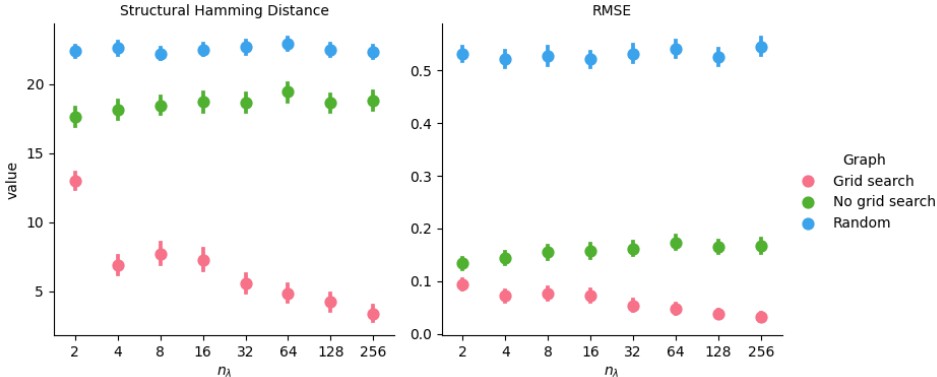

Figure 3: Average metrics for an increasing number of grid search steps ($n_\lambda$)
Reports of 'EXP2', an increasing number of grid search steps for 50 patients, 10 tasks, 10 observations per task, and random graph of mean degree 3. The predicted graph (red dots) is compared with a random graph from the same graph distribution (blue dots), and also with a predicted graph directly fitted from a random initialization with the optimal $\lambda_*$ found from grid search (green dots). Error bars represent bootstrap 95% confidence intervals.

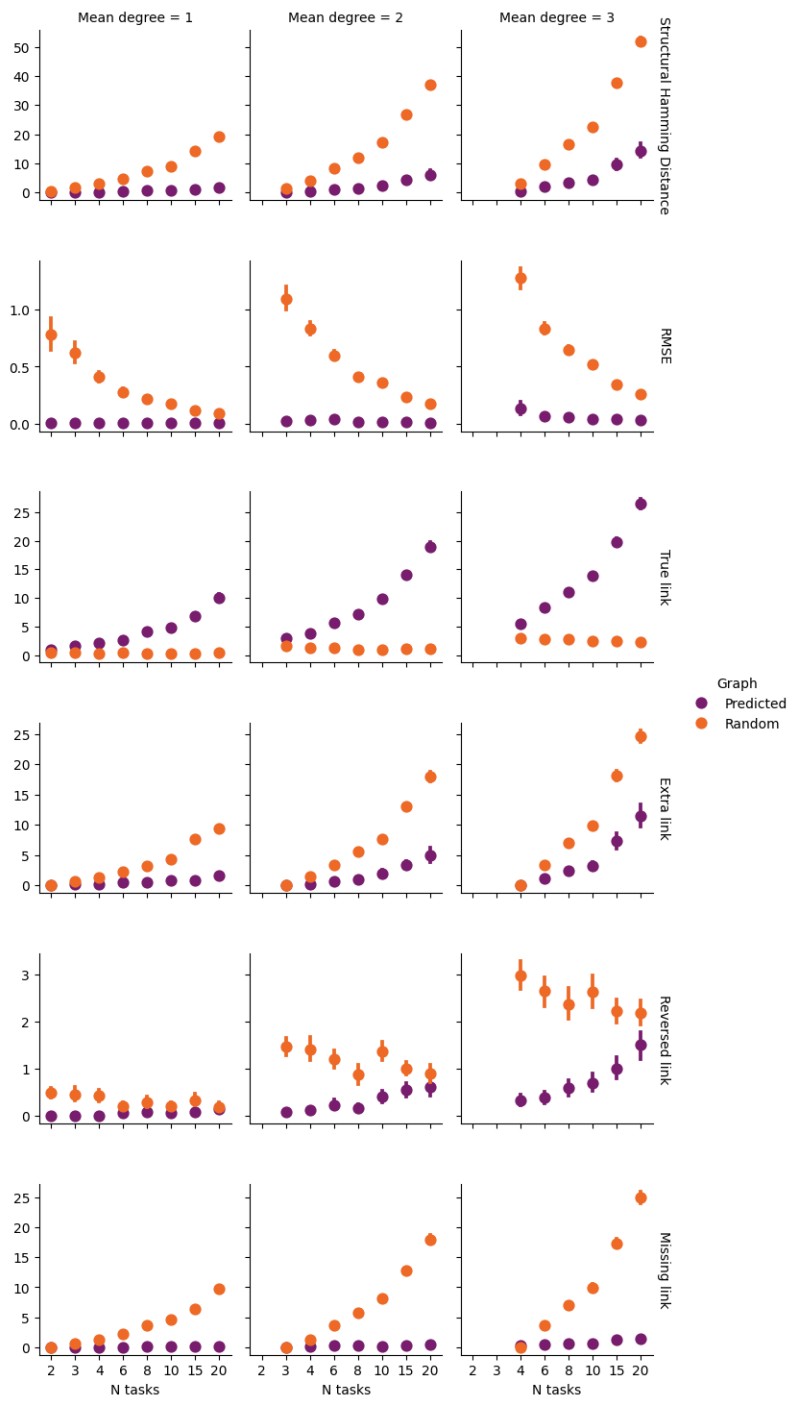

Figure 4: Average metrics for an increasing number of tasks
Reports of 'EXP3', an increasing number of tasks for 50 patients, and 10 observations per task. The predicted graph (purple dots) is compared with a random graph from the same graph distribution (orange dots). Error bars represent bootstrap 95% confidence intervals.

