# OpenReview forum: "Spectral structure learning for clinical time series"
_NeurIPS.cc/2024/Workshop/BDU — NeurIPS BDU Workshop 2024 Poster_

### Official Review · Reviewer_sr1R · 2024-09-26
**This paper introduces StructGP, a structure learning algorithm for irregularly sampled clinical time series based on a multi-task Gaussian process (GP) model.**

**Rating:** 5
**Confidence:** 4

**Review:**

-The StructGP model is specifically designed for clinical time series, which are often irregularly sampled. Many existing algorithms struggle with non-uniform data, but StructGP effectively models these challenges using continuous-time Gaussian processes.
-The incorporation of a Directed Acyclic Graph structure allows StructGP to model ordered conditional dependencies, offering potential for causal inference in time series data. This is valuable in clinical applications, where understanding cause-effect relationships can influence decision-making.
Limitations:
-While recall is strong, precision is lower at 0.71%, indicating a higher number of false positive dependencies. This can lead to overfitting in real-world clinical data where spurious connections may be identified, potentially introducing misleading conclusions.
-StructGP assumes that the observed time series capture the full dependency structure, but clinical data often contain unmeasured confounders or latent variables. The model might not be robust in the presence of such confounders, leading to biased or incorrect dependency structures.
-While the NOTEARS-based optimization is efficient for small to medium graphs (up to 20 tasks), scaling it up to larger graphs or datasets could lead to significant computational costs. The grid search for optimal sparsity parameters also adds to the computational burden.

---

### Official Review · Reviewer_DGtr · 2024-10-03
**Spectral structure learning for clinical time series**

**Rating:** 7
**Confidence:** 5

**Review:**

This is a well-written paper that proposes a novel method for learning the dependency structure of clinical time series data collected from Electronic Health Records (EHRs).

Quality:

The paper is well-structured and easy to follow.
The authors clearly define the problem and their approach.
The methodology is described in detail, including relevant formulas and references.
The evaluation methodology is sound, with simulations to assess the accuracy of the algorithm.

Clarity:

The writing is clear and concise.
The authors use appropriate technical language.
Figures and tables are informative and well-explained.

Originality:

The proposed method, StructGP, is a novel approach to learning the dependency structure of clinical time series data.
The use of a differentiable definition of acyclicity for structure learning is innovative.

Significance:

This work has the potential to improve the understanding of relationships between different clinical time series, which could lead to better diagnosis and treatment of diseases.
The method is applicable to a wide range of clinical time series data, making it potentially impactful.
Pros:

The proposed method can handle irregularly sampled data, which is common in EHRs.
The method is able to learn both the direction and strength of dependencies between time series.
The simulations show promising results in terms of accuracy.
Cons:

The method is computationally expensive due to the grid search and proximal gradient steps.
The performance of the method may be sensitive to the specific scaling of the data and the presence of unmeasured confounders.
The paper does not address how the method would scale to very large datasets.

Overall:

This is a well-written and innovative paper that proposes a promising method for learning the dependency structure of clinical time series data. Further research is needed to address the limitations mentioned above, but this work has the potential to be a valuable tool in clinical research.

---

### Decision · Program_Chairs · 2024-10-09

Accept (Poster)